# *In-silico* study of approved drugs as potential inhibitors against 3CLpro and other viral proteins of CoVID-19

Imra Aqeel [1], Abdul Majid[1], Tahani Jaser Alahmadi[2]*, Areej Althubaity [3]

1 Biomedical Informatics Research Lab, Department of Computer & Information Sciences, Pakistan Institute of Engineering & Applied Sciences, Nilore, Islamabad, Pakistan, 2 Department of Information Systems, College of Computer and Information Sciences, Princess Nourah Bint Abdulrahman University, Riyadh, Saudi Arabia, 3 Cybersecurity Department, Umm Al Qura University, Makkah, Saudi Arabia

* tjalahmadi@pnu.edu.sa

## Abstract

The global pandemic, due to the emergence of COVID-19, has created a public health crisis. It has a huge morbidity rate that was never comprehended in the recent decades. Despite numerous efforts, potent antiviral drugs are lacking. Repurposing of drugs presents a low-cost and rapid solution for finding new drugs by exploiting known drugs. In this study, we employed an integrated *In-Silico* approach using molecular docking and machine learning regression models to explore the potential inhibitors against key proteins of SARS-CoV-2. A library of 5903 drugs from the ZINC database was retrieved and screened against three crucial viral targets: *Spike* glycoprotein *(7LM9)*, *main protease 3CLpro (7JSU)*, and *Nucleocapsid protein (7DE1)*. Binding affinities were predicted by using molecular docking, and subsequent predictive regression models, Decision Tree Regression (DTR), Gradient Boosting, XGBoost, Extra Trees, KNNR, and MLP, were constructed employing MACCS molecular fingerprints. Among them, the DTR model had better predictive performance, as indicated by the highest R² and lowest RMSE. The highest ranked compounds possessed good binding affinities (−12.6 to −19.7 kcal/mol) and favorable pharmacokinetics. Importantly, five novel candidate compounds, namely ZINC003873365, ZINC085432544, ZINC008214470, ZINC085536956, and ZINC261494640, had multi-target potential and optimal binding interaction. This computational analysis yields useful information for lead prioritization and sets the stage for additional *in vitro* and *in vivo* confirmation of these drug candidates to combat COVID-19.

## 1. Introduction

According to https://www.worldometers.info/coronavirus/, as of 10 July 2024, there were over 704.75 million confirmed cases of COVID-19 worldwide and over 7.01 million deaths from the virus. This has created an unparalleled global health emergency.

**Data availability statement:** Data used in this study is already available in public repositories https://zinc20.docking.org/substances/subsets/world/ https://www.rcsb.org/structure/7JSU https://www.rcsb.org/structure/7LM9 https://www.rcsb.org/structure/7DE1

**Funding:** The funding of the Princess Nourah bint Abdulrahman University Researchers Supporting Project number (PNURSP2025R513), Princess Nourah bint Abdulrahman University, P.O. Box 84428, Riyadh 11671, Saudi Arabia.

**Competing interests:** The authors have declared that no competing interests exist.

The respiratory tract infections caused by the causative agent, Severe Acute Respiratory Syndrome Coronavirus 2 (SARS-CoV-2), range in severity from subclinical and mild common cold-like symptoms to fatal acute respiratory distress syndrome and multiple organ failure. SARS-CoV-2 is an enveloped single positive-stranded RNA virus that belongs to the Coronaviridae family [1]. This virus attacks airway epithelium cells and is spread via aerosol. The severity of the disease has decreased as a result of vaccination campaigns and rising public immunity [2]. Nonetheless, the range of drugs available for COVID-19 treatment and prevention is restricted. There is research to explore the potential inhibitors for SARS-CoV-2 infection.

Even if vaccinations have helped in the COVID-19 outbreak, their full efficacy depends on herd immunity [3]. Finding efficient antiviral drugs is a top priority due to the virus's quick mutation, diminishing immunity, and expensive vaccination. Currently, few antivirals are approved under emergency authorization that have low effectiveness against SARS-CoV-2 replication. The first approved antiviral *remdesivir*, [4] has a broad-spectrum antiviral activity with RNA polymerase inhibitor. It can treat a restricted number of hospitalized patients but requires parenteral administration. Oral nucleotide analog *molnupiravir* [5] has a broad-spectrum antiviral action. This is used clinically in individuals with mild to moderate COVID-19 patients who are at high risk of developing severe illness. Lastly, the most popular *paxlovid* consists of a combination of *ritonavir*, a *CYP3A4* inhibitor, and *nimatrelvir*, the primary protease inhibitor for SARS-CoV-2. In addition to suppressing host or viral proteases or RNA synthesis, other methods for disrupting the coronavirus life cycle include stopping the virus's entry into cells through the ACE2 receptor, preventing the assembly of new viral particles, and obstructing the virus's uptake pathway. Modulating inflammatory pathways can also have an impact on the infection's outcome [6,7]. Other already promoted antivirals like *lopinavir* or *ritonavir* finally shown fruitless in treating the disease [8]. This highlights the necessity of developing novel targeted antivirals for the treatment of SARS-CoV-2 [9]. The drugs have well-established pharmacological qualities. The repurposing reduces the time it takes to introduce a medicine [10]. In this scenario, machine learning (ML)-based computational approaches can help by eliminating the expensive trial-and-error procedures for biochemical screening [11]. For instance, *baricitinib*, a medication for rheumatoid arthritis, was repurposed using artificial intelligence approaches to treat SARS-CoV-2 infections [12]. This medication has shortened recovery times and improved clinical status in COVID-19 patients when taken in conjunction with *remdesivir* [13]. In a study [14], a number of antiviral drug classes and recently authorized oral medicines, including molnupiravir and PF-07321332, have demonstrated effectiveness against SARS-CoV-2 by inhibiting viral replication.

In the study, for molecular docking and drug screening, we selected three targeted proteins of COVID-19 including *Spike (S)*, *main protease 3CL (3CLpro)*, and *Nucleocapsid (N)* proteins. These proteins are responsible for the binding of virions with host cells. The angiotensin-converting enzyme-2 (ACE2) receptor has to attach to the host cell, and this is seen by an extremely glycosylated envelope protein known as the *Spike* protein [15,16]. This is used to bind to the ACE2 receptor and cell infection and

membrane fusion [17,18]. Additionally, the receptor binding domain (RBD) is essential to attach to the ACE2 peptidase domain. It is the primary target of deactivating the antibodies produced during infection [19,20]. The *Nucleocapsid (N)* protein is another viral protein of SARS-COV-2. It identifies the viral RNA forming a helical symmetrical structure that plays essential multiple functions in the life cycle of coronavirus [21]. It attaches itself to the virus's genomic RNA to create a ribonucleoprotein complex (RNP) [22]. The *Nucleocapsid (N)* protein has garnered a lot of interest in the creation of drugs and vaccines. On the other hand, the main protease *3CLpro (Mpro)* is essential for cleaving viral polyproteins into the functional non-structural proteins required for viral replication [23]. The *3CLpro* is considered a possible target for many COVID-19 antiviral treatments in development. The *3CL* protease is a desirable option for the creation of protease inhibitors. The creation of precise and effective *3CL protease* inhibitors is still a difficult task [24,25].

The complexity along with the fast evolution of COVID-19 poses serious challenges in identifying promising repurposed drugs as a potential treatment. Molecular docking is one of the computational approaches that has proved to be informative on potential drug candidates in the identification of possible drug candidates [26]. It allows quick screening of large compound numbers against specific targets, producing valuable information regarding their potential efficacy and binding interactions. The study focuses on the identification of promising compounds as new drugs for the treatment of COVID-19.

The main novelty of our in-silco study is that we integrated molecular docking analysis with a machine learning-based framework to systematically evaluate chemical drug candidates against three critical COVID-19 proteins: the *Main* protease (*Mpro*), the *Spike* protein, and the *Nucleocapsid* protein. From this approach, we identified five promising multi-target drug compounds effective against *Spike*, *3CLpro*, and *Nucleocapsid* proteins. Unlike previous studies, our approach automates potential candidate selection for multi-target COVID-19 proteins, helping to enhance efficiency. Additionally, we incorporated a diverse set of ML models to improve the binding affinity predictions. This enhances the reliability of binding affinity predictions and comparative analysis of model performance, which is overlooked in other studies. Furthermore, in the study, we conducted an in-depth analysis of the physicochemical properties of promising potential inhibitors. This analysis assesses the viability of drug-like characteristics essential for clinical application.

Our objective was to identify selective promising inhibitors with enhanced binding affinity for SARS-CoV-2 multi-targeted proteins. For this purpose, we performed molecular docking using well-known Autodock Vina software [27]. This software has helped to assess the potential interactions with COVID-19-related targets. We collected 5903 drug molecules from the ZINC database. The binding affinity of these drugs was calculated against the multi-target proteins including *Spike (S)*, *main protease 3CL (3CLpro)*, and *Nucleocapsid (N)* proteins. These values were employed in the drug screening process by applying an optimal threshold that reduced the initial set of compounds up to 5537. From this refined drug pool, the top five compounds were shortlisted for pharmacokinetic analysis.

In the next step, various QSAR regression models were developed by employing ML regression approaches, including Decision Tree regression (DTR), Gradient Boosting regression (GBR), Extra Trees regression (ETR), K-Nearest Neighbor regression (KNNR), Multi-Layer Perceptron regression (MLPR), and XGBoost regression (XGBR). These models were constructed and trained using the MACCS (Molecular ACCess Systems) fingerprint feature descriptor type that was computed using the PaDEL descriptor program [28]. We found these models effective in estimating binding affinities towards the multi-targeted proteins. $R^2$ and RMSE are statistical measurements that were used to evaluate the simulated outcomes of regression models. The comparative analysis revealed that the DTR model performed better than the other regression models. Finally, the pharmacokinetic analysis of the shortlisted inhibitors was conducted to assess their physiochemical properties. This *in-silico* analysis investigates the favorable behavior of selected molecules in biological systems.

The subsequent sections are organized as follows: The material and proposed ML-based framework developed are described in Section 2. The results and discussion are shown in Section 3. Section 4 presents the conclusions of our study.

## 2. Materials and methods

ML-based computational framework is shown in Fig 1. In the figure, Module A includes different steps to extract approved drugs and target proteins from their corresponding databases and compute the binding affinities between the target proteins and the extracted drug molecules. However, Module B highlights details related to the development of various regression models. Finally, Module C describes the drug analysis and molecular docking.

### 2.1. Module A: data set preparation and molecular docking

Module A consists of the data-preprocessing stages for molecular docking as follows:

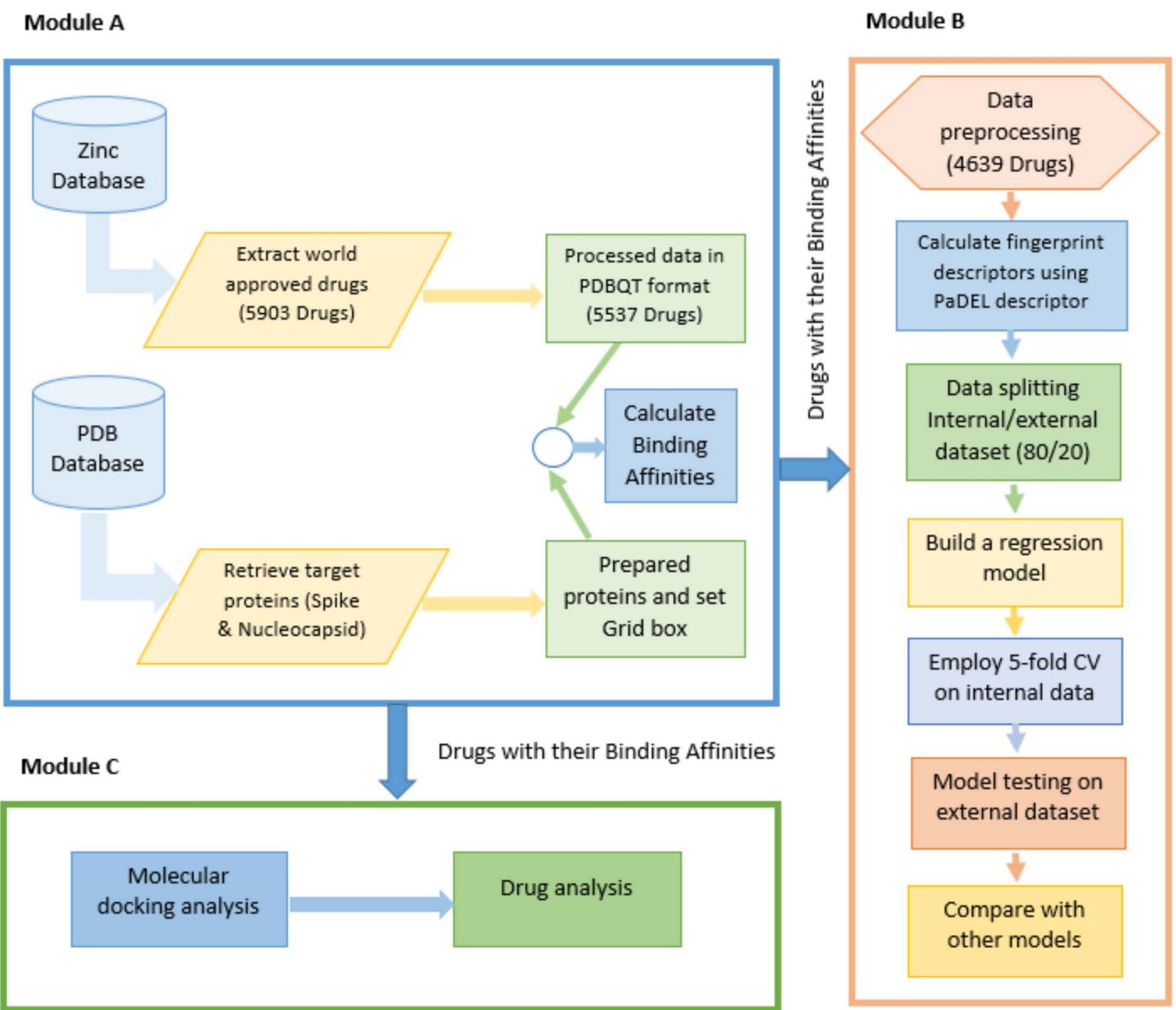

**Fig 1. Three main modules (A to C) are constructed in the proposed computational framework.**

**2.1.1. Targeting the *Spike*, *3CLpro*, and *Nucleocapsid* proteins.** The SARS-CoV-2 *Spike* protein, *3CLpro* (main protease, also called *Mpro*), and *Nucleocapsid* proteins are vital targets for therapeutic research. They are essential to the virus's capacity to infect, reproduce, and assemble inside host cells. The life cycle of SARS-CoV-2 involves the *Spike, 3CLpro,* and *Nucleocapsid* protein. These proteins also show promise as therapeutic targets for blocking the virus and halting the development of COVID-19.

The *Spike* protein correlates with changes in the transmissibility and immune-evasion capabilities of the virus. Targeting this protein might offer protection against other SARS-CoV-2 strains. *3CLpro* in the SARS-Cov-2 protease enzyme catalyzes the processing of polyproteins from the viral RNA for generating functional proteins needed to replicate and assemble the viruses. Inhibition of the activity of the *3CLpro* enzyme interrupts the replication cycle of the virus. The drugs targeting *3CLpro* may also offer broad-spectrum antiviral efficacy against other members of the coronaviruses.

The *Nucleocapsid* protein is another important component in SARS-CoV-2 that interacts with the viral RNA genome to create a ribonucleoprotein complex. This complex is essential during virus packaging and replication. The drugs may limit the reproduction of viruses if they cause the disruption of the ability of the virus to properly package its RNA or build new virions by targeting the *Nucleocapsid* protein. Further, the *Nucleocapsid* protein plays an important role in regulating the host's immune response, and its inhibition may enhance the host's immunological power to counter the virus. Since it is less prone to mutation compared with the *Spike* protein, this protein acts as a stable target for the synthesis of antiviral drugs.

**2.1.2. Data set.** The Zinc database was used to extract the drug candidates approved by the FDA [29]. Initially, a data set including 5903 drugs was collected from https://zinc20.docking.org/ on 2 October 2023. This database contains more than 1.4 billion compounds. The Zinc database is openly accessible to the general population.

**2.1.3. Data preprocessing.** The Zinc database, which consists of initially 5903 approved drugs in SMILES format, was stepped through several conversion steps for further analysis. The SMILES strings were converted into SDF format using the OpenBabel-2.4.1 software [30]. The files in format SDF were converted into PDBQT format, which enabled the use of the drugs for calculating their binding affinity with the target proteins. The files that could not be converted were excluded from the data set. After this processing, the data set was reduced to 5537 drugs.

**2.1.4. Visualization of molecular docking.** The crystal structures of the *Spike* (PDB ID: *7LM9*), *3CLpro* (PDB ID: *7JSU*), and *Nucleocapsid* (PDB ID: *7DE1*) proteins were downloaded from the RCSB Protein Data Bank on 2 October 2023. Ligands from each of the structures were stripped to optimize the protein models. Water molecules and alternative side chains were removed from each structure; however, polar hydrogen atoms were retained. Macromolecules were added with Kollman charges. A grid box of 30×30×30 and spacing of 1 Å were used to include the active sites of these proteins. The centers of the x, y, and z coordinates for *7LM9*, *7JSU*, and *7DE1* were placed at 32.951, −13.678, −11.595; −11.046, 12.826, 67.749; and 29.275, 18.899, 17.009, respectively. Molecular docking was carried out using AutoDock Vina version 1.2.0 with default settings. Ligands were prepared in PDBQT format using OpenBabel software. The binding affinities were calculated in kcal/mol. In all cases, the interaction with the lowest binding energy was the most favorable pose for the ligand binding. Fig 2 shows the crystal structures of *7LM9*, *7JSU*, and *7DE1* proteins with resolutions of 1.53 Å, 1.83 Å, and 2.00 Å, respectively.

In this research, we used the crystal structure of the SARS-CoV-2 main protease (3CLpro) with PDB ID: 7JSU to conduct molecular docking analysis. This structure exists in its monomeric state in the database, but it has an annotation as a biological assembly in the form of a homodimer (Global Stoichiometry: Homo 2-mer – A2), (https://www.rcsb.org/structure/7JSU) which shows the enzyme's active form. We chose to use the monomeric form because it was available in high resolution and many previous virtual screening studies had used it. It's worth noting that AutoDock Vina, the docking tool we used in our analysis works on individual binding pockets and doesn't simulate protein–protein interactions, which makes docking on monomeric structures a sound method. Also, the ligand in 7JSU forms a covalent bond with Cys145, which supports the use of the monomeric model to evaluate inhibitory potential. Earlier studies have shown that the active

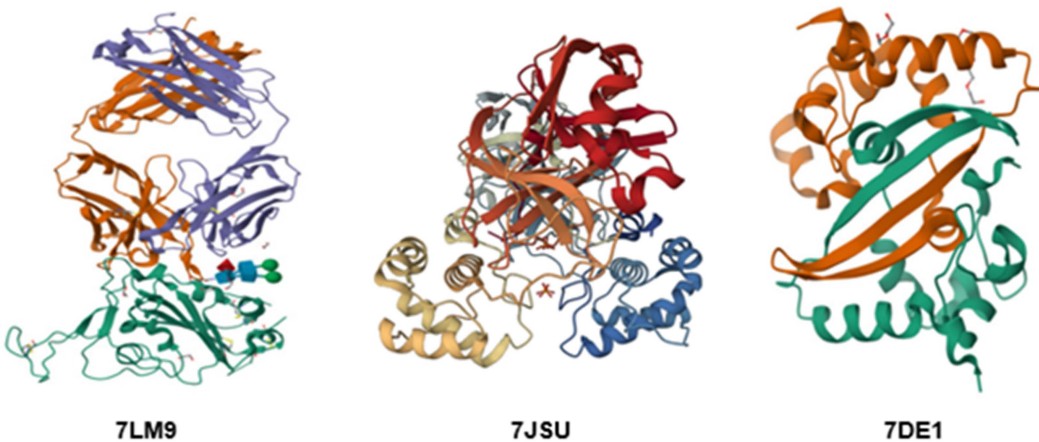

**Fig 2. Crystal structures of Spike *7LM9*, 3CLpro *7JSU*, and Nucleocapsid *7DE1*.**

binding site of 3CLpro the catalytic dyad (His41 and Cys145) is within a single protomer, which allows for valid ligand interaction analysis without the immediate need for the dimeric form [31]. Many computational drug discovery efforts [32,33] have used monomeric 3CLpro to initial screen and identify leads, which backs up the relevance and adequacy of this approach.

However, the processes of dimerization contribute to the functions of the enzyme and might also affect the stability and movement of the active site [34]. Some of the research studies [35,36] indicated that binding a ligand might influence dimerization as well as the activity of the enzyme. To enhance our docking experiment or widen the range of the structural scope we, too, included the dimeric form of the nucleocapsid (N) protein C terminal domain, PDB ID: 7DE1, in the dimer analysis. This structure has also been shown to exhibit homodimeric stoichiometry (A2) alongside the experimental evidence and the PDB biological assembly. The addition of this dimeric structure illustrates our consideration for the oligomeric states and how fundamental these states are in binding dynamics. Therefore, we approached the problem by integrating computations on structures simplified to their monomeric and dimeric representations which are considered where biologically relevant using geometrical reasoning placed in a contextual framework called virtual screening. This *in silico* stepwise progression of research design described above verges from monomer-based screening into dimer validation balances computational feasibility with biological relevance during the preliminary stages of drug development.

Using several protein structures, we investigated the binding of possible inhibitors taking into account both covalent and non-covalent interactions in this research. Covalent ligand (UED) was used for docking research with 7JSU, which represents the monomeric form of the SARS-CoV-2 main protease (3CLpro); we also included non-covalent docking to expand the range of our models. Particularly, 7DE1, which represents the C-terminal domain (CTD) of the SARS-CoV-2 nucleocapsid protein, was used to investigate non-covalent interactions in its dimer form. For study of protein-protein interactions and possible non-covalent inhibitor binding at the dimer interface, this structure is pertinent. Furthermore 7LM9, a complex of an antibody fragment (Fab) with the receptor-binding domain (RBD) of the SARS-CoV-2 spike protein. The interactions here are first and foremost non-covalent, including hydrogen bonding, hydrophobic interactions, and electrostatic forces. Together with the 7JSU monomer, these non-covalent protein structures allow a thorough assessment of covalent and non-covalent inhibitors hence ensuring applicability of our computational models for various drug discovery methods including reversible inhibitors and covalent binders.

## 2.2. Module B: QSAR modeling

In this module, we will first explain the data preprocessing and feature extraction steps. Then, we will explain how regression models are developed to predict the binding affinity of drugs using their molecular descriptors. Finally, we will present the performance evaluation measures.

**2.2.1. Data cleaning.** In order to train the regression models, the input data set is cleaned to remove all the duplication. The drugs with missing values of binding affinity are filtered from the data set. In the final data set, a total of 4639 drugs were used for model development. The elimination of redundant data and exclusion of those drugs without values of their binding affinity ensures the accuracy and integrity of the data set.

**2.2.2. Feature extraction.** A vector of fingerprint descriptors represented molecular components of the drug compound. We used a built-in PaDEL- Descriptor tool for the standardization of tautomer and desalting. The MACCS fingerprint descriptor was used to predict the binding affinities of drug molecules. It encodes the presence or absence of 166 pre-specified chemical features or substructures in the form of a binary vector, where each single bit accounts for a specific feature. The details of categories of features in the MACCS fingerprint is tabulated in Table 1.

The MACCS fingerprint describes the structural characteristics of the chemical compounds in a compressed form. So, it is useful for predicting binding affinity in the drug discovery process. It gives a binary value to check whether certain substructures, such as functional groups, aromatic rings, HBD, HBA, or types of bonds, are present within a molecule or not. The MACCS fingerprints allow the efficient comparison of similarities between a query molecule and known ligands with

**Table 1. Categories of features in the MACCS fingerprint.**

| Sr. No | Category Name | Features |
|---|---|---|
| 1 | Atom and Element Types | • Presence of specific atoms like carbon, oxygen, nitrogen, sulfur, phosphorus, halogens (F, Cl, Br, I).<br>• Atom types such as sp2 or sp3 hybridized carbons. |
| 2 | Functional Groups | • Alcohols (OH groups)<br>• Amines (primary, secondary, tertiary)<br>• Carbonyl groups (ketones, aldehydes)<br>• Carboxyl groups (COOH)<br>• Ethers and esters<br>• Halogen groups (fluoro, chloro, etc.) |
| 3 | Bond Types and Patterns | • Single, double, triple bonds<br>• Aromatic bonds<br>• Conjugated double bonds |
| 4 | Rings and Aromaticity | • Presence of aromatic rings (e.g., benzene).<br>• Heteroaromatic rings (with N, O, or S).<br>• Aliphatic rings (non-aromatic cyclic structures). |
| 5 | Hydrogen Bonding Features | • Hydrogen bond donors (HBD) such as hydroxyl or amine groups.<br>• Hydrogen bond acceptors (HBA) like oxygen or nitrogen in carbonyls and ethers. |
| 6 | Molecular Complexity and Size-Related Features | • Presence of heavy atoms (atoms other than hydrogen).<br>• Atom count thresholds (e.g., more than 5 or 10 non-hydrogen atoms).<br>• Presence of small aliphatic or aromatic fragments. |
| 7 | Presence of Heteroatoms | • Structures containing non-carbon atoms such as nitrogen, oxygen, sulfur, or phosphorus.<br>• Patterns involving heteroatom-substituted rings. |
| 8 | Topological Features | • Simple chain length patterns (e.g., long or short aliphatic chains).<br>• Number of branches or terminal atoms.<br>• Substituents on aromatic or heteroaromatic rings |

established binding profiles in the prediction of binding affinity. We can find compounds that share structural properties with ligands known to bind efficiently to a particular target protein by comparing these fingerprints with similarity metrics.

The similar binding behavior of molecules with structural similarity makes this similarity-based approach a useful proxy for estimating binding affinity. The MACCS fingerprint allows the comparison useful for virtual screening provides the ability to analyze vast chemical libraries to find potential candidates with favorable binding affinities. The MACCS fingerprints provide an efficient way to identify potential binders.

**2.2.3. Regression model development/implementation.** For regression model development, we have selected several algorithms including DTR, GBR, ETR, XGBR, MLPR, and KNNR. These algorithms are extensively used in various applications of biomedical research. Here, we will explain briefly with the implementation point of view. These algorithms have their own strengths that make it possible to effectively develop the complex relationships between molecular descriptors and binding affinity. For example, DTR model non-linear patterns using the hierarchical decision-making processes. The DTR model [37] resembles a flow chart with nodes, branches, and leaves. The interior nodes of the decision tree represent a test for a particular attribute; the branch shows the test outcome, and each leaf node shows a prediction or class label. For a DTR, the value of a leaf node is the continuous value of target values in training samples associated with the same leaf node. GBR and XGBR models employ ensemble learning for improving predictive performance by combining several weak models. These models improve prediction by reducing bias and enhancing generalization through iterative corrections. ETR model is a variation of a random forest and improves generalization by generating multiple decision trees with randomized data partitions reducing over-fitting. However, the distance-based KNNR model is useful in detecting local patterns. This model can capture local structure-activity relationships. However, the MLPR model has flexibility in modeling complex patterns, while incorporating robustness to their predictions.

Decision Tree Regression (DTR) is more effective than models like GBR, XBGR, KNNR, ETR and MLPR. This is because it is better in the detection of non-linear correlations between features/independent variables and binding affinities. It is not subject to any distributional assumption on the data compared to parametric models, so it is more flexible. It chooses relevant features, minimizing over-fitting and maximizing efficiency, as opposed to KNNR, which does not perform well with high-dimensional data. Furthermore, DTR is less sensitive to outliers and computationally more efficient compared to ensemble-based approaches such as GBR, ETR and XGBR. All these strengths position DTR as a promising contender for drug repurposing binding affinity prediction.

During training, the optimal hyper-parameters of these regressors are used to adjust the bias and variance trade-off. This strategy makes models more reliable and robust performance. The best combination of parameters and hyper-parameters for each model was obtained while improving predictive and avoiding over-fitting. Table 2 provides different values of hyper-parameters for the construction of prediction models. This table shows, that for DTR and ETR models, we adjusted the optimal values of tree depth, minimum samples per split, maximum features, and number of estimators. During training, DTR model divide the feature space into subsets recursively based on the optimization of the feature space while minimizing the variance reduction of the target variable. The KNNR model was optimized by setting the

**Table 2. Details of hyper-parameters setting of predictive models.**

| Sr. No | Model Name | Hyper-Parameters |
|---|---|---|
| 1 | DTR | max_depth = "none", min_sample_split = 3, max_features = "auto" |
| 2 | ETR | max_depth = "none", min_sample_split = 2, n_estimators = 100, max_features = 'auto' |
| 3 | KNNR | n_neighbors = 1, weights = "distance" |
| 4 | GBR | n_estimators = 200, learning_rate = 0.1, max_features = 'auto' |
| 5 | XGBR | n_estimators = 1000, learning_rate = 0.1, max_features = 'auto', lambda = 1, alpha = 0 |
| 6 | MLPR | hidden_layer_sizes = (100,50), activation = 'tanh', solver = 'adam', max_iter = 1000 |

number of nearest neighbors. GBR and XGBR were fine-tuned using different number of estimators and their learning rate. Finally, optimal parameters/hyper-parameters values are given in the table. The λ and α parameters of XGBR model help balance the trade-off between bias and variance, and their tuning is essential for building an optimal model. For λ = 1 means moderate L2 regularization that is applied to control model complexity by shrinking large coefficients. For α = 0 means no L1 regularization, so no feature elimination is enforced, and all features contribute. MLPR was fine-tuned by adjusting the hidden layers and their sizes, activation function, solver, and maximum iteration.

These regression models are developed to predict the binding affinity with high accuracy. As mentioned in Section 2.2.1, the input data set contains a total of 4639 drug compounds. This data set has an 80:20 ratio between its internal and external data sets. We employed 5-fold cross-validation on internal data set. The MACCS fingerprint molecular descriptor described in Section 2.2.2 is employed as the feature set for this purpose. The model's performance is evaluated using the external data set. The prediction performance of models is reported in terms of $Q^2$, $R^2$, and RMSE measures.

**2.2.4. Evaluation metrics.** The performance of regression models is reported using coefficient of determination ($R^2$), predictive squared correlation coefficient ($Q^2$), and root mean square error (RMSE). The $R^2$ value measures the proportion of variance in the dependent variable explained by the independent. The $Q^2$ measure indicates how well a model predicts on testing data. A higher $Q^2$ value highlights better generalizability of the model on unseen data. In regression modeling, $R^2$ and $Q^2$ measures are useful for assessing the fitting and predictive power of the model. The difference between $R^2$ and $Q^2$ values should be less than 0.3 [38]. A model performs acceptably in regression with a $Q^2$ score of 0.5, whereas a score above 0.9 indicates outstanding performance. Higher values indicate better model performance with the range between 0 and 1. However, the RMSE measure represents the relative error between the actual and predicted binding affinity. The lower values indicate better model performance.

### 2.3. Module C: Molecular docking and drug analysis

In module C, molecular docking and drug analysis are carried out, explained as follows.

**2.3.1. Molecular docking analysis.** Molecular docking was carried out using AutoDock Vina. The protein structure and ligand were used to predict the most favorable binding orientations and interactions within the protein's active site. From this analysis, we obtained the binding affinity, expressed as the free energy change (ΔG) in kcal/mol. The lower binding energy values indicate the stronger and more stable ligand-protein interactions. However, the lowest energy pose is considered the optimal binding mode. AutoDock Vina is employed to generate multiple binding poses for the ligand and rank their energy scores. The top-ranked poses are the focus of further investigation. In addition, the consistency of projected ligand poses in relation to a reference structure or experimentally acquired data is evaluated by computing the root mean square deviation (RMSD), where lower RMSD values give more reliable docking predictions.

We have shown how the ligand fits into the active site and interacts with the residues around it. The visualization of the binding site contributes to a more profound understanding. This would help to determine the orientation of the ligand and possible binding pockets. The docking score obtained through AutoDock Vina is ranked according to their binding affinities. It is possible to select the best position for further investigation. The docking analysis gives useful information regarding the ligands-protein interaction.

**2.3.2. Drug analysis.** While considering drug repurposing *in-silico* research to explore whether licensed drugs could be used for novel therapeutic uses, a thorough physicochemical evaluation is required. Such factors are crucial in defining the drug's permeability, solubility, bioavailability, and pharmacokinetic behavior. The molecular weight is an important physicochemical factor that affects a drug's overall absorption and its ability to penetrate biological membranes. The affinity of the drug for the lipid versus the aqueous environment, or lipophilicity, is also indicated as logP and impacts its solubility and distribution. The count of HBD and HBA determines the aqueous solubility of the drug and its ability to interact with the biological target.

Some examples of such additional features that explain the structural complexity of the drug and its ability to be involved in hydrophobic interactions include the existence of rings and the presence of heavy atoms. The presence of rings and heavy atoms are other features that contribute to the drug's structural complexity and its capacity to engage in hydrophobic interactions. These features often enhance the drug's interaction with target proteins. The presence of heteroatoms, other than carbon and hydrogen, provides insight into a drug's potential to engage in hydrogen bonding and other electrostatic interactions. Additionally, analyzing the proportion of sp3 hybridized carbons can help to estimate the three-dimensionality of drugs and their possible impact on binding affinity and metabolic stability. A higher SP3 proportion typically suggests the increased molecular complexity and potentially leads to improved drug-target interactions and better pharmacokinetic profiles. Finally, it can predict the pharmacokinetic behavior, safety, and efficacy of repurposed pharmaceuticals, which makes evaluation easier for new therapeutic applications.

## 3. Results and discussion

In this section, we will first describe the comparative performance of several regression models in terms of $R^2$ and RMSE measures. Next, we will explain the possible inhibitors and binding affinities for the multi-target proteins *Spike*, *3CLpro*, and *Nucleocapsid*. Using the MACCS fingerprint feature set, we evaluate how well our proposed DTR model predicts binding affinities. Following that, we will explain the outcomes of the molecular docking studies of approved drugs and discuss their interactions with multi-target proteins. Finally, we will analyze the physicochemical properties of the drug compounds to assess their effectiveness against the multi-target proteins.

### 3.1. Evaluation of the QSAR model

This study will identify the best regression model based on Decision Tree Regression (DTR). A detailed explanation of the 4639 drug compounds used in the development of the model is given in Section 2.2.1. In Section 2.2.2, a detailed description of the MACCS fingerprint is provided. These features are used to develop regression models. The effectiveness of the DTR model in predicting binding affinities is measured across *Spike*, *3CLpro*, and *Nucleocapsid*. How much variance in the predictions of binding affinities can be explained by the model is represented by the $R^2$ measure. The model obtained the best $R^2$ values for 0.95 for *Spike*, 0.97 for *3CLpro*, and 0.95 for *Nucleocapsid*. This shows the improved performance for each target. The regression plot of the best DTR model is shown in Fig 3. This figure illustrates the relationship between the actual and predicted binding affinities for multi-target proteins of *Spike*, *3CLpro*, and *Nucleocapsid*. This demonstrates that the DTR model explains most of the variance in the predictions.

The performance of the best DTR model is further evaluated using MSE and RMSE metrics. According to both MSE and RMSE measures, the *3CLpro* obtained the best accuracy among the three proteins, with an MSE of 2.46 and an RMSE of 1.57. Similarly, for the *Spike* protein, this model achieved an MSE of 2.76 and an RMSE of 1.66. In comparison to *Spike*, for *Nucleocapsid* the protein model shows slightly better prediction performance, with the lowest MSE of 2.21 and RMSE of 1.49. This demonstrates the reliability and robustness of the DTR model in predicting binding affinities using MACCS fingerprints. This model performs well for all proteins with minimal error margins. Table 3 shows the performance of the best DTR model using $Q^2$ and $R^2$ measures. The details related to these measures are provided in Section 2.2.4. According to this table, the DTR model produced a $Q^2$ value of 0.80 and an $R^2$ value of 0.95 for the *Spike* protein. However, for *3CLpro* and *Nucleocapsid* proteins, the corresponding values are 0.77, 0.97 for *3CLpro* and 0.77 and 0.93 for $Q^2$ and $R^2$ measures, respectively. This highlights a small difference between $Q^2$ and $R^2$ measures in the range of 0.15–0.20. This shows that the DTR model is effective in predicting the binding affinity of drug molecules with *Spike*, *3CLpro*, and *Nucleocapsid* proteins.

 

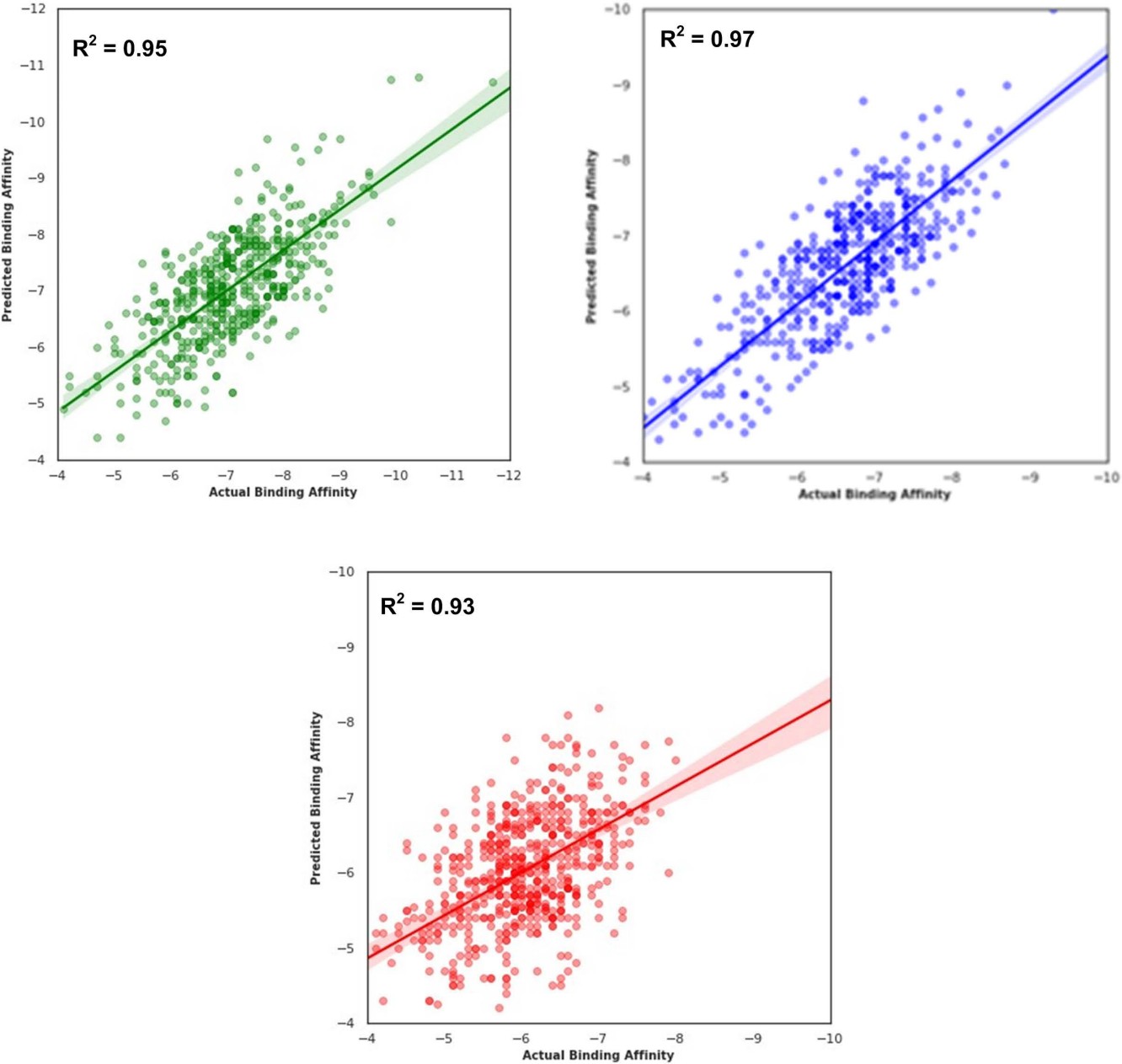

**Fig 3. Regression plots of R2 using MACCS fingerprint features.**

## 3.2. Comparative analysis

In this section, we will compare the predictive performance of ETR, KNNR, GBR, MLPR, XGBR, and DTR models using RMSE and $R^2$ measures. Table 4 presents a comparison in predicting the affinities of binding for three major proteins associated with SARS-CoV-2: *Spike*, *3CLpro*, and *Nucleocapsid*.

The DTR regression model exhibited the best predictive performance with $R^2$ values of 0.95, 0.97, and 0.93 for *Spike, 3CLpro*, and *Nucleocapsid*, respectively. This model has also obtained the lowest RMSE values of 1.66, 1.57, and 1.49.

**Table 3. Performance of the DTR model for the MACCS fingerprint.**

| Sr. No | Protein Name | Q² | R² | Difference |
|---|---|---|---|---|
| 1 | Spike | 0.80 | 0.95 | 0.15 |
| 2 | 3CLpro | 0.77 | 0.97 | 0.20 |
| 3 | Nucleocapsid | 0.77 | 0.93 | 0.16 |

**Table 4. Performance comparison of regression models using the MACCS fingerprint.**

| Regression Model | Spike | | 3CLpro | | Nucleocapsid | |
|---|---|---|---|---|---|---|
| | R² | RMSE | R² | RMSE | R² | RMSE |
| DTR | 0.95 | 1.66 | 0.97 | 1.57 | 0.93 | 1.49 |
| ETR | 0.79 | 1.80 | 0.85 | 1.80 | 0.70 | 1.63 |
| KNNR | 0.66 | 1.82 | 0.96 | 1.86 | 0.87 | 1.71 |
| GBR | 0.70 | 1.84 | 0.88 | 1.81 | 0.69 | 1.62 |
| MLPR | 0.61 | 1.74 | 0.60 | 1.64 | 0.65 | 1.61 |
| XGBR | 0.52 | 1.85 | 0.54 | 1.85 | 0.52 | 1.73 |

This indicates that the DTR model explains the largest proportion of variance in the binding affinity predictions. It has the most accurate predictions among the other models. The ETR model also performs relatively well, particularly for *3CLpro*, with an $R^2$ of 0.85 and an RMSE of 1.80. However, its performance decreases for *Spike* ($R^2 = 0.79$) and *Nucleocapsid* ($R^2 = 0.70$), with RMSE values slightly higher at 1.80 and 1.63, respectively.

The KNNR model achieves good results for *3CLpro* with an $R^2$ value of 0.96 and an RMSE value of 1.86. However, it shows lower predictive performance for *Spike* ($R^2 = 0.66$) and *Nucleocapsid* ($R^2 = 0.87$), with RMSE values of 1.82 and 1.71. The GBR model displays moderate performance, with $R^2$ values of 0.70 for *Spike*, 0.88 for *3CLpro*, 0.69 for *Nucleocapsid*, and RMSE values slightly higher than those of the KNNR model.

The MLPR and XGBR models obtained lower $R^2$ values ranging from 0.52 to 0.65 across all proteins and higher RMSE values than other models. XGBR model has the lowest predictive ability for three proteins with $R^2$ values of 0.52 for both *Spike* and *Nucleocapsid* and 0.54 for *3CLpro*.

Overall, the results indicate that DTR outperforms the other regression models in predicting binding affinities across all three proteins, with the lowest RMSE and the highest $R^2$ values. We inferred that the DTR model is the most suitable for the task of predicting binding affinity using MACCS fingerprints, among other models.

### 3.3. Molecular docking

In the study, we want to evaluate how effectively the selected drug compounds interacted with three important target proteins. For this purpose, molecular docking was employed to predict and analyze the interactions between the ligands and target proteins. This technique helps in understanding the orientation and binding affinity of ligands within the active sites of the proteins. We estimated the binding affinities of drug compounds retrieved from the Zinc database using a ligand-based docking technique. After the drug compounds were formatted in PDBQT, the binding affinities between compounds and the target proteins were calculated in kcal/mol.

Fig 4a demonstrates the 3D interaction image of the target protein *7LM9* complex with the ligand *ZINC003873365* attached. Similarly, Fig 4b shows the 3D interaction of the target protein *7JSU* complex with the same ligand. Lastly, the 3D interaction view of the target protein *7DE1* complex with the same ligand *ZINC003873365* is shown in Fig 4c.

The *7LM9* protein binding pocket harbors ACE2-interacting residues like Tyr449, Gln493, and Asn501 that are pivotal in host cell recognition. The active site of *7JSU* protein is constituted by His41 and Cys145 of the catalytic dyad, and

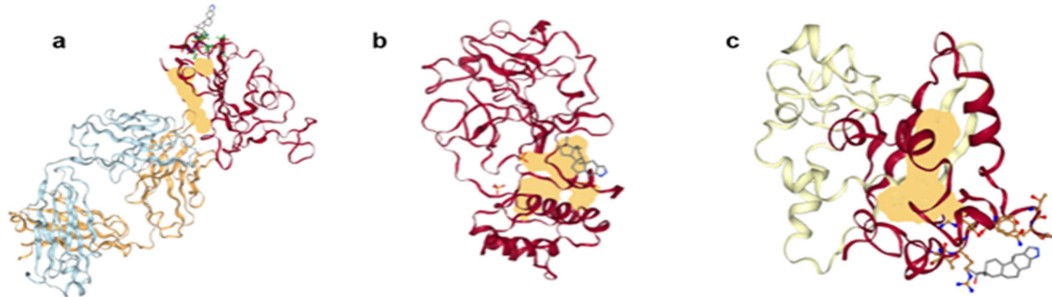

**Fig 4. 3D interaction view of proteins, (a) 7LM9, (b) 7JSU, (c) 7DE1, with ligand ZINC03873365.**

important residues like Glu166, Phe140, and His163 that play a part in substrate recognition. The S1 pocket is hydrophilic, while hydrophobic interactions are facilitated by S2 and S4 pockets. The *7DE1* protein binding pocket, the RNA-binding domain (RBD), with the RNA-binding groove and the most critical residues (Arg92, Tyr109, and Phe115) responsible for stabilizing viral genome packaging. Docking is to search for compounds that interfere with RNA-protein interactions.

In the study, we reported the performance of target proteins *7LM9*, *7JSU*, and *7DE1* protein using drug score and simple score measures. These measures (https://proteins.plus/ access on 25 July 2024) are used to evaluate proteins for drug purposes and other biotechnological applications. The drug score measure assesses the ability of a protein to be targeted by small molecules based on protein structural features that favor drug targeting, drug similarity, and binding affinity. A higher drug score indicates a greater likelihood of the protein being a viable therapeutic target. However, the simple score provides a more direct assessment of the protein's general properties, taking into account hydrophobicity, molecular weight, and content of particular amino acids. For example, in industrial processes or enzyme activity, a high simple score demonstrates suitable characteristics for functionality and performance in different applications. These scores give valuable information for the scientists who work in the field of drug development and protein modification.

Table 5 presents the three target proteins (*7LM9, 7JSU,* and *7DE1*) and their potential binding pocket based on the 3D structure of the proteins. This table gives information for assessing the suitability of these proteins for drug targeting. These calculations are got by using the webserver (https://proteins.plus/).

This table highlights the most suitable binding pocket P_0 for each protein and various other parameters such as pocket volume ($Å^3$) and surface area ($Å^2$) providing an insight into the size and accessibility of these binding sites. The binding pocket volume is expressed in cubic angstroms ($Å^3$), which quantifies the potential capacity for bound ligands. For example, pocket volume 535.49 $Å^3$ of *7LM9* protein that binds ligands with high affinity. Another important measure of understanding interaction sites is the solvent-accessible surface area, measured in ($Å^2$). *7LM9* protein has a surface area of 591.53 $Å^2$, providing a good region for ligand binding. The drug score (0.87) is very potent binding. Thus, *7LM9* protein stands best from the rest with a high drug score. This indicates promising drug-binding capabilities despite the lower simple score. The *7DE1* protein has a drug score of 0.74, and the *7JSU* protein has a drug score of 0.82. This shows the greater possibilities for favorable interactions. Moreover, for a simple score, we checked for the basic functional properties

**Table 5. Drug Score of target proteins.**

| Protein Name | Pocket Name | Volume $Å^3$ | Surface $Å^2$ | Drug Score | Simple Score |
|---|---|---|---|---|---|
| 7LM9 | P_0 | 535.49 | 591.53 | 0.87 | 0.35 |
| 7JSU | P_0 | 746.24 | 1257.1 | 0.82 | 0.54 |
| 7DE1 | P_0 | 525.18 | 161.17 | 0.74 | 0.38 |

of the binding pockets. For *7LM9*, *7JSU*, and *7DE1* proteins, the simple scores were 0.35, 0.54, and 0.38, respectively, giving a little idea about what makes them unique. These results demonstrate the advantages of each protein in ligand binding, thus making them highly suitable for drug discovery projects. The drug score reflects the overall drugability of a binding pocket, integrating variables such as binding affinity, dimensionality, and molecular compatibility. The scores of 1.0 indicate a more favorable binding site. However, the simple score shows the ligand compatibility and its binding strength in the pocket. These scores can highlight target proteins for further study, especially *7LM9* and *7JSU* proteins, which have obtained higher drug scores. This indicate these proteins have a better chance that their binding pockets can be used for drug repurposing.

The root mean square deviation (RMSD) of the ligand is measured from the natural site of the protein complex. This measure is utilized to assess the accuracy of the docking procedure. A ligand molecule with a better docking geometry has a lower RMSD value. Our study shows that in their ideal positions, each of the five ligands had an RMSD value of zero. This suggests that the docking geometry is highly accurate. The binding affinities (BA) values between the five drug compounds that score highest, and the associated proteins, are displayed in Table 6.

In the study [30], *Griseofulvin* and its derivatives were evaluated against COVID-19 targets using molecular docking with (*CID 144564153*) and (*CID 46844082*) compounds. Their docking scores demonstrate the structural stability of *Griseofulvin CID 441140*. Results showed the docking score (−6.8 kcal/mol) with *Mpro*, while its derivative *CID 144564153* was most potent at −9.49 kcal/mol, followed by *CID 46844082* at −8.44 kcal/mol against *Mpro* and *ACE2*. However, we identified five promising ligands, ZINC003873365, ZINC085432544, ZINC085536956, ZINC008214470, and ZINC261494640, with the best binding affinities to three critical COVID-19 proteins: *7LM9*, *7JSU*, and *7DE1*. It is to be noted that ZINC003873365 showed superior binding, with BA values of −19.7, −15.1, and −19.2 kcal/mol for *7LM9, 7JSU,* and *7DE1*, respectively. The study of ligand poses and the associated binding energies contribute to the understanding of the connection between the ligand's molecular structure and binding affinity. These drug molecules have a strong connection, indicating a promising target.

Fig 5 represents the 2D interaction view of optimal pose of five top-ranked drug compounds interacting with the target proteins *7JSU*, *7LM9* and *7DE1* proteins. The best binding affinities for these drug compounds range from −19.7 to −12.6 kcal/mol. The precise orientation and conformation of the ligand molecule that results in the lowest binding affinity with the target protease is referred to as the "optimal pose." The ideal ligand posture with respect to the target is indicated by the binding energy value that is the highest negative. Van Der Waals, conventional hydrogen bonds, carbon hydrogen bonds, Pi-cations, Pi-sulfur, Pi-Pi T-shaped, unfavorable donor-donor, alkyl, and Pi-alkyl interactions are all depicted in 2D view of the protein-ligand interaction. The overall stability of protein-ligand binding is significantly influenced by these many kinds of interactions.

Among many of non-covalent forces, hydrophobic interaction and hydrogen bonding are particularly important in stabilizing the ligand inside the active site in protein–ligand interactions. Nonpolar regions of the ligand and protein avoid water, hence pushing them together and causing hydrophobic interactions. By helping good van der Waals forces and lowering the entropic cost of solvation, these interactions greatly affect the binding affinity. Conversely, hydrogen bonds provide directionality and specificity, typically serving as major factors in ligand identification. Typical hydrogen bonds include an

**Table 6. Top-ranked five ligands with target proteins.**

| Sr. No. | Zinc ID | 7LM9 (BA) (kcal/mol) | 7JSU (BA) (kcal/mol) | 7DE1 (BA) (kcal/mol) |
|---|---|---|---|---|
| 1 | ZINC003873365 | −19.7 | −15.1 | −19.2 |
| 2 | ZINC085432544 | −15.3 | −14.4 | −14.0 |
| 3 | ZINC085536956 | −14.4 | −13.9 | −12.6 |
| 4 | ZINC008214470 | −15.3 | −13.6 | −14.3 |
| 5 | ZINC261494640 | −13.9 | −13.6 | −14.4 |

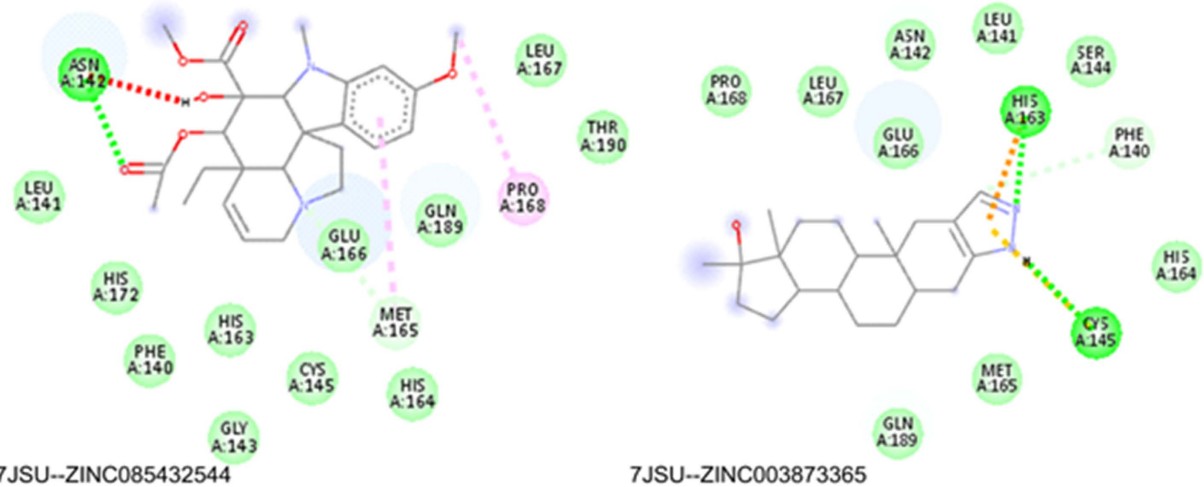

7JSU--ZINC085432544

7JSU--ZINC003873365

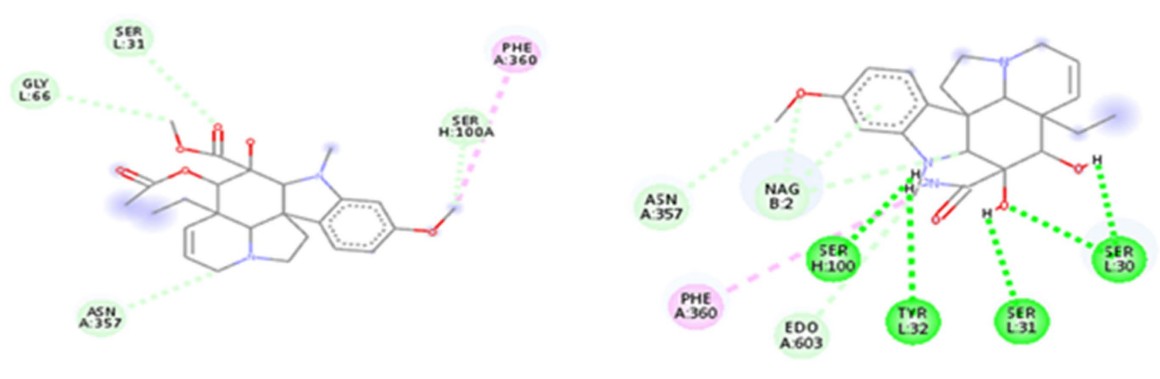

7LM9--ZINC085536956

7LM9--ZINC008214470

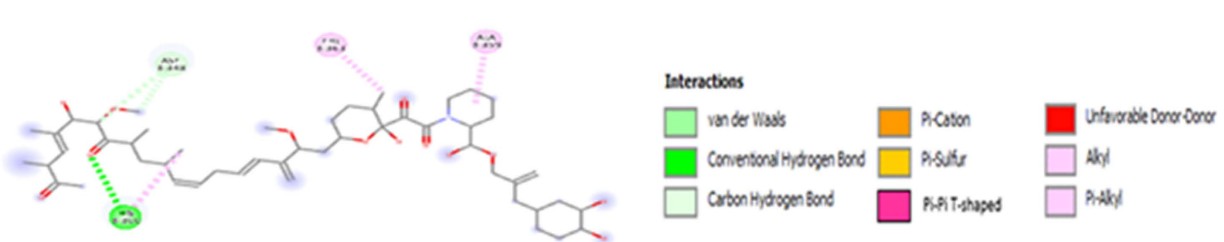

7DE1--ZINC261494640

**Fig 5. 2D interaction view of top ranked ligands with 7JSU, 7LM9 and 7DE1 Proteins.**

electronegative donor and acceptor with bond distances ranging from 2.5 Å to 3.5 Å and bond angles preferably above 120°, both of which are essential for maximum binding strength. The detail of hydrophobic interaction and H-bond is given in the supplementary file S1 Table.

Strong hydrogen bonding and hydrophobic interactions with important active site residues supported the favorable binding affinities of the topranked ligands (such as ZINC085432544, ZINC003873365, ZINC085536956, ZINC261494640 and ZINC008214470) across several SARS-CoV-2 target proteins as discovered in silico. The interaction analysis indicates that including more polar functional groups, especially those able to form directional hydrogen bonds, could still increase binding specificity and strength. Moreover, improving the packaging within the binding pocket would entail maximizing the spatial orientation of hydrophobic groups. These structural insights offer a useful foundation for prospective compound optimization work, thus directing the design of more powerful and selective antiviral candidates.

### 3.4. Physiochemical analysis of drug candidates

From the previous molecular docking study, we identified the five most promising drug compounds with higher potency and robust interactions with three target proteins, *Spike*, *3CLpro*, and *Nucleocapsid.* The supplementary file S2 Table provides a description of these five drug compounds, including Zinc IDs, molecular formulas, SMILES (Simplified Molecular Input Line Entry System) format, and their important geometrical and 2D structural features.

Table 7 provides some useful physicochemical properties of five potential therapeutic drug compounds, including heavy atoms, heteroatoms, rings, sp3 fraction, molecular weight (Mol.Wt), partition coefficient (LogP), hydrogen bond donors (HBD), and hydrogen bond acceptors (HBA). These properties highlight the efficiency and bioactivity of medicinal drugs. We employed these properties to assess the effectiveness of potential drug compounds in biological systems. We provide a detailed description of the impacts of these physicochemical properties on the selected drug compounds as follows:

The molecular weight is regarded as one of the most critical criteria for the assessment of pharmacokinetics and pharmacodynamics properties of a drug compound. The drug compounds have to penetrate through cell membranes and interact with target proteins. The optimum molecular weight increases the likelihood of binding with the target proteins. This table shows that the molecular weights vary widely in size from 328.5 g/mol to 900.1 g/mol. We observed that heavier heteroatoms in the selected drugs are associated with their larger molecular weights. The bigger molecules might bind to targets more successfully. From this table, we observed that ZINC003873365 has a lower molecular weight (328.5 g/mol) but a higher sp3 fraction of 0.86. This might be associated with a higher level of 3D complexity and better binding interactions. In that perspective, it is better than ZINC261494640, which has a similar sp3 fraction (0.74) but is associated with a higher molecular weight ((900.1 g/mol).

The logP is another factor that has an important role in impacting the therapeutic binding affinity. The higher logP value implies higher lipophilicity and a high tendency to interact with the hydrophobic domains of proteins. The logP has been identified to be the key descriptor responsible for molecule therapeutics binding to their target [39]. This table shows that values of logP vary from 2.732 to 5.527, i.e., the logP values of drugs vary in the range of 3 and 5. The drug development relies on the lipophilicity (logP). It affects drug absorption, distribution, metabolism, and excretion (ADME)

**Table 7. Physiochemical properties of selected drug compounds.**

| Sr. No | Zinc ID | Mol. Wt (g/mol) | LogP | Rings | Heavy Atoms | Hetero Atoms | HBD, HBA | Fraction sp3 |
|---|---|---|---|---|---|---|---|---|
| 1 | ZINC085432544 | 810.9 | 3.99 | 9 | 59 | 13 | 5, 10 | 0.59 |
| 2 | ZINC003873365 | 328.5 | 4.118 | 5 | 24 | 3 | 2, 2 | 0.86 |
| 3 | ZINC085536956 | 778.9 | 4.754 | 9 | 57 | 12 | 4, 9 | 0.53 |
| 4 | ZINC261494640 | 900.1 | 5.527 | 4 | 64 | 14 | 4, 10 | 0.74 |
| 5 | ZINC008214470 | 753.9 | 2.732 | 9 | 55 | 12 | 7, 8 | 0.58 |

characteristics. The lipophilicity of a drug influences its absorption, distribution, metabolism, and excretion (ADME) properties, which are directly related to logP value. Among the selected compounds, ZINC261494640 has the highest logP value 5.527, indicating its higher lipophilicity providing strong potential for higher absorption and bioavailability. On the other hand, ZINC085536956 has a relatively lower logP of 4.754, reflecting lower lipophilicity than previous. In contrast, ZINC008214470 exhibits the lowest lipophilicity with a logP value of 2.732, indicating lower absorption and membrane permeability compared to the other compounds. The remaining compounds, ZINC003873365 (logP = 4.118) and ZINC085432544 (logP = 3.99), fall within the moderate range, indicating balanced lipophilic properties suitable for drug development. This variation in lipophilicity directly impacts the drug's pharmacokinetic behavior. We can assess how well drugs can interact with biological membranes and are distributed throughout the body.

The number of rings and heavy atoms are important characteristics that influence the binding affinity of particular targets. Very large or too many rings may interfere with binding or increase toxicity [40]. Table 7 indicate that potential molecules have rings from 4 to 9 and heavy atoms from 24 to 64. This provides the structural complexity of the selected compounds, which can significantly influence the stability, potency, and selectivity of the drug compounds. The molecules with a greater number of heavy atoms and rings are more complex. This will increase the interaction with the target receptor to enhance binding affinity. ZINC261494640, with 64 heavy atoms, is the most complex, which could lead to stronger selective interactions with the receptor. Similarly, both ZINC085432544 and ZINC085536956 (59 and 57 heavy atoms) indicate potential higher interaction with the target protein. In contrast, ZINC003873365, with 24 heavy atoms, is a less complex structure with fewer interactions but possibly better solubility and faster absorption. However, ZINC008214470, with 55 heavy atoms, keeps a balance between complexity and potential interaction. Overall, the number of rings and heavy atoms within these potential compounds plays a critical role in determining their pharmacological profiles. The complex structures generally have enhanced stability and binding properties. From this table, we observe that heteroatom count varies from 3 to 14. They may give clues to the potential binding interactions that might occur between drug molecules and their specific target. Heteroatoms of nitrogen and oxygen atoms may form hydrogen bonds or other electrostatic interactions with the target residues.

HBD and HBA are essential for drug-protein interactions. The higher values contribute to the formation of more drug-protein interactions. This table shows that potential drugs have HBD ≤ 5 and HBA ≤ 10, satisfying Lipinski's Rule of Five, except ZINC008214470 with an HBD value of 7. The increasing HBD and HBA values enhance the capacity of the drug molecule to create hydrogen bonds with the binding site of the targeted protein. This may lead to higher binding affinity. Among the compounds, ZINC008214470 has the highest HBD value of 7. This may result in stronger hydrogen bonding interactions but could potentially reduce membrane permeability, affecting its absorption. However, ZINC003873365 has the lowest HBD and HBA values of 2, indicating less hydrogen bonding, which could enhance membrane permeability and lead to better absorption and bioavailability. In contrast, ZINC085432544 and ZINC261494640 have the highest HBA values of 10, which may increase their potential for interactions with biological molecules. This could also impact their solubility and permeability. However, the ZINC085536956 compound presents balanced HBD and HBA values of 4 and 9, respectively. This indicates a moderate hydrogen bonding potential that may have good balance between solubility and permeability.

The sp3 fraction is an important indicator of carbon atoms with three or more single bonds. The higher sp3 fractions is associated with the increased water solubility, whereas low levels may result in poor bioavailability or decreased target selectivity [41]. From this table, we observe that fraction sp3 has a range between 0.53 and 0.86, representing the percentage of carbon atoms in the drug that are sp3 hybridized. Most drugs have a sp3 fraction value above 0.50. A drug whose sp3 value is significantly higher will exhibit more 3D character and is more effective when binding to a particular target. Overall, these physiochemical properties are crucial for assessing the effectiveness of the potential drug compounds in biological systems, influencing their absorption, distribution, metabolism, and excretion characteristics.

In this study, we demonstrated the physicochemical characteristics of the drugs while repurposing. Table 8 lists the generic name and original prescription of our short-listed drugs for repurpose against COVID-19.

**Table 8. Short-listed drugs for repurposing in this study.**

| Sr. No. | Zinc ID | Generic Name | Original Prescription | New Indication |
|---|---|---|---|---|
| 1 | ZINC003873365 | Stanozolol | Certain forms of breast cancer, anemia, and hereditary angioedema (HAE). | COVID-19 |
| 2 | ZINC085432544 | Vinblastine | Breast cancer, neuroblastoma, Hodgkin's and non-Hodgkins lymphoma, testicular cancer, histiocytosis, mycosis fungoides, and Kaposi's sarcoma. | COVID-19 |
| 3 | ZINC085536956 | Vinorelbine tartrate | Advanced or metastatic non-small cell lung cancer. | COVID-19 |
| 4 | ZINC008214470 | Vindesine | Malignant lymphoma, Acute leukaemia, Hodgkin's disease, acute erythraemia, and acute panmyelosis. | COVID-19 |
| 5 | ZINC261494640 | 41-O-demethyl rapamycin | Prevent organ rejection in kidney transplant patients, tumor-based cancers, and coat stents implanted in heart disease patients. | COVID-19 |

The top-ranked identified ligands (ZINC003873365, ZINC085432544, ZINC085536956, ZINC008214470, and ZINC261494640) are mostly anticancer drugs. Although their antiviral effects have not been directly determined, their high affinities in binding to SARS-CoV-2 proteins (Spike, 3CLpro, and Nucleocapsid) indicate repurposing potentials. Most anticancer drugs display broad-spectrum activity, such as antiviral, based on mechanisms including kinase inhibition, immune modulation, or apoptosis induction. Further *in vitro* and *in vivo* work is required to establish whether the compounds have direct antiviral activity against SARS-CoV-2.

## 4. Conclusion

In this study, we have identified potential drugs that could be repurposed for the treatment of COVID-19 by checking the possible target proteins, *Spike*, *3CLpro*, and *Nucleocapsid*. Using a computational approach that can integrate molecular docking and machine learning, we were able to identify five drugs that have high binding energies ranging between −19.7 and −12.6 kcal/mol. Our proposed DTR regression model has produced better predictions of binding affinities among other regression methods with improved $R^2$ values (0.95, 0.97, and 0.93) and RMSE values (1.66, 1.57, and 1.49) for the *Spike*, *3CLpro,* and *Nucleocapsid* proteins, respectively.

We also carried out the physicochemical properties of selected compounds to demonstrate their action mechanisms in the biological environment. This has determined their efficacy and safety profile. Our results showed these compounds are suitable potent inhibitors of multi-target proteins associated with COVID-19. This study provides a sound basis for further trials in order to prove the validity of these compounds as potential COVID-19 therapeutics.

Follow-up research, including *in vitro* and *in vivo* studies, is necessary to confirm their effectiveness. Moreover, exploring the possible synergistic effects of these compounds in combination with other treatments could yield valuable insights. Our methodology could also be applied to other viral diseases and conditions beyond virology, making drug repurposing a promising strategy. This study highlights the significance of utilizing computational tools to expedite the drug discovery process, particularly during emerging pandemics and urgent health challenges. The scope of this study is limited to *in-silico*. However, in the future, we intend to validate these computational findings through experimental studies, including *in vitro* assays to assess binding affinity and inhibitory activity, as well as *in vivo* studies to evaluate pharmacokinetics.

## Supporting information

**S1 Table. Hydrophobic and H bonding interaction of top ranked ligands with 7JSU, 7LM9 and 7DE1 proteins.** (PDF)

**S2 Table. Description of the top-ranked five drug compounds.**
(PDF)

## Acknowledgments

The authors extend their appreciation to the Princess Nourah bint Abdulrahman University Researchers Supporting Project number (PNURSP2025R513), Princess Nourah bint Abdulrahman University, Riyadh, Saudi Arabia.

## Author contributions

**Conceptualization:** Imra Aqeel.

**Data curation:** Imra Aqeel.

**Formal analysis:** Abdul Majid, Tahani Jaser Alahmadi, Areej Althubaity.

**Funding acquisition:** Tahani Jaser Alahmadi.

**Methodology:** Imra Aqeel.

**Resources:** Abdul Majid, Tahani Jaser Alahmadi.

**Software:** Imra Aqeel.

**Supervision:** Imra Aqeel, Abdul Majid.

**Validation:** Imra Aqeel, Areej Althubaity.

**Visualization:** Tahani Jaser Alahmadi.

**Writing – original draft:** Imra Aqeel.

**Writing – review & editing:** Imra Aqeel, Abdul Majid, Areej Althubaity.

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
