## [Decision Letter · Decision Letter 0]

PONE-D-25-07472To explore the potential inhibitors against multi-target proteins of COVID-19 using in-silico studyPLOS ONE

Dear Dr. Aqeel,

Thank you for submitting your manuscript to PLOS ONE. After careful consideration, we feel that it has merit but does not fully meet PLOS ONE’s publication criteria as it currently stands. Therefore, we invite you to submit a revised version of the manuscript that addresses the points raised during the review process.

We look forward to receiving your revised manuscript.

Kind regards,

Ahmed A. Al-Karmalawy, PhD

Academic Editor

PLOS ONE

“The authors extend their appreciation to the Princess Nourah bint Abdulrahman University Researchers Supporting Project number (PNURSP2023R384), Princess Nourah bint Abdulrahman University, P.O. Box 84428, Riyadh 11671, Saudi Arabia.”

Reviewers' comments:

Reviewer's Responses to Questions

**Comments to the Author**

1. Is the manuscript technically sound, and do the data support the conclusions?

Reviewer #1: Yes

Reviewer #2: Yes

Reviewer #3: Partly

2. Has the statistical analysis been performed appropriately and rigorously? 

Reviewer #1: Yes

Reviewer #2: N/A

Reviewer #3: I Don't Know

3. Have the authors made all data underlying the findings in their manuscript fully available?

Reviewer #1: Yes

Reviewer #2: Yes

Reviewer #3: Yes

4. Is the manuscript presented in an intelligible fashion and written in standard English?

Reviewer #1: Yes

Reviewer #2: Yes

Reviewer #3: Yes

5. Review Comments to the Author

Reviewer #1: The authors provided a computational approach for identifying promising inhibitors, of natural origin, against multiple SARS-CoV-2 biotargets. They adopted combined molecular docking and machine learning regression approaches to identify potential shits. This manuscript is relevant, valuable in the field of drug discovery. However, there are some points should be considered prior publication.

1. The authors adopted the PDB.file (7JSU) for performing the computational analysis which is deposited in its monomeric state. Typically, the 3CLpro is a homodimer relating to its biological activity (10.1126 / science.abb3405) and so computational study should have been performed in its homodimeric state since it is the active form of the protein. Additionally, a comparative data analysis of each 3CLpro protomer could be provided for gaining greater insights regarding the effect of ligand binding on dimerization since the dimerization state is proximity with the 3CLpro canonical binding site. It is advised that the authors would discuss this, while elaborating on the adequacy of computational screening on the 3CLpro dimer versus monomer states.

https://doi.org/10.1038/s41598-021-88630-9

https://doi.org/10.1080/07391102.2021.1880481

2. The co-crystalline ligand (UED; N~2~-[(benzyloxy)carbonyl]-N-{(2S)-1-hydroxy-3-[(3S)-2-oxopyrrolidin-3-yl]propan-2-yl}-L-leucinamide) at the 7JSU PDB.file has been reported as covalent inhibitor for the virus protein. This Michael acceptor inhibitor was used by the authors for developing models that used to screen libraries of reversible acting drugs. It is recommended to validate the obtained models on other PDB.file with reversible non-covalent 3CLpro inhibitor (e.g. PDB ID: 7L0D).

1. Within the molecular modelling studies, the authors are advised to provide more details on the identified target topology and pocket description prior to presenting the docking findings.

2. The authors should elaborate more on the ligand-target interaction patterns. Ligand-residue interactions should be annotated in terms of both the bond distances and angles. Especially for Hydrogen bonding, this type of compound-protein polar interaction should be presented within hydrogen bond distances as well as bond angles since hydrogen bond depend on both. Authors should mention the Hydrogen bond angles as well as their distances, since the strength of hydrogen bonding is based on both parameters in a way to ensure the adequacy of optimum hydrogen bonding.

3. It is advised that the authors provided a MM_PB(GB)SA free binding energy calculations with dissected energy terms (ΔGelectrostatic, ΔGvdw, ΔGsolvation,…..) to highlight the dominant potential for guiding further compound optimization and development. Identifying the hotspot residues through binding energy contribution is also advised. FastDRH webserver is feasible to perform such approach (http://cadd.zju.edu.cn/fastdrh/).

4. Based on the study results, what are the take-away messages. Authors are advised to highlight the suggested structural modifications that would improve the compound’s affinity based on the in silico findings. These insights would be beneficial for guiding future lead optimization and development.

Reviewer #2: 1.Describe the cross-validation methodology and machine learning model hyperparameter selection.

2. Explain the superior performance of Decision Tree Regression (DTR) over alternative models.

3. Provide a list of the top-ranked ligands along with their characteristics in an additional file.

4. Offer in vivo or in vitro tests as a means of verification.

5. Indicate whether any of the medication candidates that were found had antiviral properties.

6. Verify that every figure has the appropriate label and has all relevant information (such as R2 values in regression graphs).

Reviewer #3: Dear Dr. Imra Aqeel et al.,

Regarding the manuscript titled (To explore the potential inhibitors against multi-target proteins of CoVID-19 using in-silico study); the following comments should be amended accordingly:

1- The title should be more descriptive or comprehensive title that fully reflects the scope and depth of your study.

2- The abstract is well-written overall, but by tightening up some sentences and providing a bit more detail on the methodology and results, you can enhance its clarity and impact.

3- Try to improve your introduction and compare your results to others like (https://doi.org/10.2147/DDDT.S354841).

4- Write in silico, in vitro, and in vivo words in italic.

5- Overall, the article is well-structured and easy to follow. The introduction provides a clear context for the research, and the progression through various sections is logical and cohesive. However, some sections could benefit from clearer transitions. For instance, the section on in silico approaches could have more detailed explanations about the specific algorithms and software used in the studies, which would provide readers with a deeper understanding of the methodologies.

6- Molecular dynamics simulation (for at least 100 ns) for the most active five complexes, compared to the co-crystallized inhibitor of each target receptor should be per formed to confirm the docking results.

7- Some tables (like Tables 7, 8, and 9) should be moved to the Supplementary Data.

8- A graphical abstract is recommended.

9- The article fails to adequately discuss the importance of experimental validation following in silico predictions. While in silico techniques are powerful, the review does not highlight how real-world testing (e.g., in vitro or in vivo experiments) is essential to confirm the efficacy and safety of the identified inhibitors. The reliance on computational data alone can be problematic, and without a proper discussion of the need for follow-up experiments, the paper might give the impression that in silico studies are sufficient on their own.

10- Improve the resolution of the 2D structures in table7, so that the chemical structure, such as bond angles and atom labels are clear even when printed or zoomed in.

11- There are some grammatical and typographical error, please revise it.

6. PLOS authors have the option to publish the peer review history of their article (what does this mean? ). If published, this will include your full peer review and any attached files.

**Do you want your identity to be public for this peer review?** For information about this choice, including consent withdrawal, please see our Privacy Policy .

Reviewer #1: **Yes**

Reviewer #2: No

Reviewer #3: **Yes**

---

## [Author Response · Author response to Decision Letter 1]

30 Apr 2025

Response to reviewers file is attached for point to point response.

---

## [Decision Letter · Decision Letter 1]

In-silico study of approved drugs as potential inhibitors against 3CLpro and other viral proteins of Covid-19

PONE-D-25-07472R1

Dear Dr. Aqeel,

We’re pleased to inform you that your manuscript has been judged scientifically suitable for publication and will be formally accepted for publication once it meets all outstanding technical requirements.

Kind regards,

Ahmed A. Al-Karmalawy, PhD

Academic Editor

PLOS ONE

Reviewers' comments:

Reviewer's Responses to Questions

**Comments to the Author**

1. If the authors have adequately addressed your comments raised in a previous round of review and you feel that this manuscript is now acceptable for publication, you may indicate that here to bypass the “Comments to the Author” section, enter your conflict of interest statement in the “Confidential to Editor” section, and submit your "Accept" recommendation.

Reviewer #2: All comments have been addressed

Reviewer #3: All comments have been addressed

2. Is the manuscript technically sound, and do the data support the conclusions?

Reviewer #2: Yes

Reviewer #3: Yes

3. Has the statistical analysis been performed appropriately and rigorously? 

Reviewer #2: N/A

Reviewer #3: Yes

4. Have the authors made all data underlying the findings in their manuscript fully available?

Reviewer #2: Yes

Reviewer #3: Yes

5. Is the manuscript presented in an intelligible fashion and written in standard English?

Reviewer #2: Yes

Reviewer #3: Yes

6. Review Comments to the Author

Reviewer #2: I have reviewed the revised manuscript, and I confirm that the author has adequately addressed all my previous comments. I have no further concerns, and I recommend the manuscript for acceptance in its current form.

Reviewer #3: This is a well written and thoroughly researched paper that makes a valuable contribution to the field.

7. PLOS authors have the option to publish the peer review history of their article (what does this mean? ). If published, this will include your full peer review and any attached files.

**Do you want your identity to be public for this peer review?** For information about this choice, including consent withdrawal, please see our Privacy Policy .

Reviewer #2: **Yes: **

Reviewer #3: **Yes: **

---

## [Editor Report · Acceptance letter]

PONE-D-25-07472R1

PLOS ONE

Dear Dr. Alahmadi,

I'm pleased to inform you that your manuscript has been deemed suitable for publication in PLOS ONE. Congratulations! Your manuscript is now being handed over to our production team.

Kind regards,

on behalf of

Associate Professor Ahmed A. Al-Karmalawy

Academic Editor

PLOS ONE